# Extract of *Juniperus indica Bertol* Synergizes with Cisplatin to Inhibit Oral Cancer Cell Growth via Repression of Cell Cycle Progression and Activation of the Caspase Cascade

**DOI:** 10.3390/molecules25122746

**Published:** 2020-06-13

**Authors:** Xiao-Fan Huang, Kai-Fu Chang, Shan-Chih Lee, Chia-Yu Li, Hung-Hsiu Liao, Ming-Chang Hsieh, Nu-Man Tsai

**Affiliations:** 1Institute of Medicine, Chung Shan Medical University, Taichung 40201, Taiwan; s9870509@gmail.com (X.-F.H.); kfchang1015@gmail.com (K.-F.C.); 2Department of Medical Laboratory and Biotechnology, Chung Shan Medical University, Taichung 40201, Taiwan; s65874@yahoo.com.tw; 3Department of Medical Imaging and Radiological Sciences, Chung Shan Medical University, Taichung 40201, Taiwan; sclee@csmu.edu.tw; 4Department of Medical Imaging, Chung Shan Medical University Hospital, Taichung 40201, Taiwan; 5Department of Life and Death, Nanhua University, Chiayi 62249, Taiwan,; joyce@nhu.edu.tw; 6Clinical Laboratory, Chung Shan Medical University Hospital, Taichung 40201, Taiwan

**Keywords:** oral cancer, cisplatin, synergism, cell apoptosis, *Juniperus indica Bertol*

## Abstract

Oral cancer—a type of head and neck cancer—is estimated to be the fifth most common cancer in Taiwan. However, efficacious therapies for oral cancer are still lacking due to drug resistance and recurrence. Consequently, the identification of new anticancer agents for clinical treatment is needed. *Juniperus indica Bertol* is a plant of the *Juniperus* genus often used as a treatment in traditional medicine due to its anti-inflammatory, antibacterial and diuretic functions. The biofunctions of *Juniperus indica Bertol* including its anticancer potential, have not been fully explored. As a result, the aim of this research was to investigate the anticancer activity of *Juniperus indica Bertol* extract (JIB extract) and determine whether JIB extract has synergistic effects with cisplatin in oral cancer. These results are the first to demonstrate that JIB extract exhibits anticancer capacity and synergizes with cisplatin to treat oral cancer. Our findings indicate that JIB extract has a potential to develop anticancer agent and chemo therapeutic adjuvant for oral cancer.

## 1. Introduction

Oral cancer is the most common type of head-neck cancer; the Ministry of Health and Welfare has estimated that among cancers, oral cancer had the fifth highest incidence in 2016 in Taiwan. Furthermore, according to the National Cancer Institute, oral cancer accounted for 2.9% and 1.6% of the incidence and mortality of all cancers, respectively, in 2017 in the United States. The five-year survival rate of patients with oral cancer decreases significantly with increasing stage, ranging from 77.7% to 32.8%, and the survival rate drops to 10% owing to recurrence and metastasis [1,2,3]. The common therapeutic procedure in oral cancer is first the administration of curable treatments, such as surgery, for early stage oral cancer and palliative treatments, including radiotherapy, chemotherapy, target therapy and combination treatments, for nonsurgical or advanced cases of oral cancer [4].The current treatment not clearly improves survival benefits of patients due to the drug resistance, tolerance to chemo-drug toxicity and high recurrent rate. Therefore, development of combination therapy represents a major new trend for oral cancer treatment [5].

Cisplatin is a platinum-based chemotherapeutic drug approved by the FDA for the treatment of advanced ovarian, testicular and bladder cancers in 1978. Several researchers have recently reported that cisplatin is also used in the treatment of various additional cancers, such as cervical cancer, lung cancer, osteosarcoma and neuroblastoma [6]. Although cisplatin displays good therapeutic efficiency in patients, it still causes severe adverse side effects, such as nephrotoxicity, peripheral neuropathy, nausea, vomiting and nephrotoxicity, which is especially lethal due to renal failure [7]. In addition. the recurrence rate of oral cancer within three years is estimated to be approximately 86%. In addition. these tumors are highly metastatic and drug resistant, resulting in noneffective treatment and poor prognosis [1]. Therefore, the need to find an anticancer agent with low toxicity that can ameliorate tumor recurrence is urgent.

The use of conventional medicines combined with complementary and alternative medicines is increasing worldwide [8,9]. Furthermore, herbal compounds have shown the potential to synergize with antitumoral drugs to boost therapeutic efficacy and relieve adverse effects [10,11]. *Juniperus* is a genus of evergreen trees that includes the four species *Juniperus viz. Juniperus communis*, *Juniperus indica Berto.*, *Juniperus recurva* and *Juniperus squamata*. Among these plants, the various biofunctions of *Juniperus communis* have been fully studied [12,13,14,15]. *Juniperus indica Bertol* also called black Juniper [16] was reported to have antimicrobial activity [17] and cytotoxic activity against brine shrimp eggs [18]. However, the anticancer activity of *Juniperus indica Bertol* extract (JIB extract) against oral cancer is still not fully understood. Hence, the purpose of the study was to investigate the anticancer potential and synergistic effect of JIB extract in combination with cisplatin to provide alternative treatment for clinicians and patients. We first examined the antiproliferative activity of JIB extract, as it was not yet clear. Our results showed that JIB extract not only inhibited oral cancer cell growth and the induction of apoptosis, but also synergized with cisplatin to downregulate the Akt/mTOR pathway and reduce cell regrowth. Consequently, JIB extract may be utilized as an adjuvant in combination with cisplatin and provide a new therapeutic strategy for oral cancer treatment.

## 2. Results

### 2.1. Inhibitory Effect of JIB Extract on Oral Cancer Cell Growth

To investigate the antiproliferative activity of JIB extract against oral cancer cells, cells were treated with JIB extract for the indicated intervals, and MTT assays were performed. OECM-1 cell viability was reduced by approximately 80% following treatment with JIB extract at concentrations of 50–200 μg/mL. The cell viability of SAS and SCC-25 cells was reduced by approximately 50–60% following treatment with JIB extract at high concentrations ranging from 50 μg/mL to 200 μg/mL (Figure 1A). *Juniperus indica Bertol* crude extract inhibited oral cancer cell growth in a dose-dependent manner. The IC_50_ of JIB extract in the oral cancer cell lines ranged from 38.12 ± 0.75 μg/mL to 65.9 ± 3.93 μg/mL (Table 1). As shown in Figure 1B, cisplatin efficiently inhibited oral cancer cell proliferation in a dose- and time-dependent manner, and its IC_50_ values ranged from 1.19 ± 0.01 μg/mL to 25.08 ± 1.53 μg/mL (Table 1). To test the cytotoxicity of JIB extract on normal cells, MDCK, a kidney epithelial cell line and SVEC cells, an endothelial cell line, were used. The MDCK and SVEC cells were slightly inhibited about 10% of cell growth by JIB extract at a concentration of 50 μg/mL, suggesting that JIB extract is less cytotoxic to normal cells (Figure 1C). As shown in Table 1, the IC_50_ of JIB extract in the normal cell lines ranged from 73.63 ± 0.9 μg/mL to 102.59 ± 14.26 μg/mL. Then, cisplatin exerted marked cytotoxic effects against normal cells, and its IC_50_ values ranged from 1.07 ± 0.25 μg/mL to 6.08 ± 0.88 μg/mL, indicating the occurrence of previously reported side effects, such as renal toxicity (Figure 1C). Next, to assess the ability of JIB extract to select tumor and normal cells, its selective index (SI) was calculated. The SI of the JIB extract in all cell lines was greater than 1 and ranged from 1.12 to 2.65; however, the SI of cisplatin in most cell lines was less than 1 and ranged from 0.18 to 0.96 (Figure 1D). In summary, JIB extract at a concentration of 50 μg/mL inhibited oral cancer cell growth and showed little cytotoxicity to normal cells. In addition, JIB extract showed greater selective ability than cisplatin.

### 2.2. Synergistic Effect of JIB Extract Plus Cisplatin on OECM-1 Cells

The former results indicated that OECM-1 cells were the most sensitive to JIB extract treatment; hence, OECM-1 cells were used in the follow-up experiments. To determine whether JIB extract plus cisplatin would exert a synergistic effect on OECM-1 cells, an MTT assay was performed to determine the inhibitory effect of the treatments. OECM-1 cells were treated with JIB extract and cisplatin for the indicated time intervals. Cells were treated with cisplatin (3 μg/mL) combined with JIB extract (0–80 μg/mL); the cell viability was reduced by approximately 20–30% when cells were treated with JIB extract at a concentration of 10 or 20 μg/mL compared with treatment with JIB extract only (Figure 2). Conversely, in OECM-1 cells treated with JIB extract (30 μg/mL) in combination with cisplatin (0–4 μg/mL), the cell viability was diminished by approximately 20–60% when the concentration of cisplatin ranged from 0.25 to 2 μg/mL in comparison with treatment with cisplatin only (Figure 2). Then, the combination index (CI) was evaluated, and a CI value less than 1 indicated synergy. JIB extract plus cisplatin had a synergistic effect, and the CI values were 0.75, 0.60 and 0.42 at 24, 48 and 72 h, respectively.

### 2.3. Obstruction Cell Cycle Progression of JIB Extract and Combinational Treatment

According to the above data, we chose an appropriate concentration of JIB extract (30 μg/mL) that had a synergistic effect with cisplatin (3 μg/mL). To fully explore the anticancer mechanisms of JIB extract plus cisplatin, we first wondered whether this combination would affect the cell cycle progression of OECM-1 cells treated for different intervals. After treatment, the cell cycle distribution was analyzed, the results of which are shown in Figure 3A,B. JIB extract induced significant cell cycle arrest at G_0_/G_1_ phase, with the proportion of cells at this stage ranging from 55.22 ± 0.46% to 74.38 ± 0.47%, and reduced the proportion of cells at the S and G_2_/M phases. Cisplatin initially blocked cell cycle progression at G_0_/G_1_ phase and decreased the proportion of cells at G_2_/M phase, which ranged from 27.76 ± 0.23% to 44.81 ± 0.59% following treatment with cisplatin for 6 to 48 h. Following combination treatment, the proportion of cells arrested at S phase ranged from 17.02 ± 0.31% to 22.99 ± 1.60% with 0–12 h of treatment, after which the proportion of the cell population at G2/M phase increased, ranging from 27.76 ± 0.23% to 40.75 ± 0.29% for 0–48 h of treatment, in contrast to the reduction in the proportion of cells at G_0_/G_1_ phase (Figure 3B). These results indicated that JIB extract or cisplatin only induced cell cycle arrest at G_0_/G_1_ phase or G_2_/M phase, respectively, while JIB extract plus cisplatin strongly induced cell cycle arrest at G_2_/M phase. As a result, we explored regulators of the cell cycle that may be affected by JIB extract and cisplatin. As shown in Figure 4A, the Western blot results showed that treatment with JIB extract or cisplatin alone reduced the protein expression of Rb, p-Rb, cdk2, cdk4, cyclin D and cyclin B, resulting in cell cycle arrest at G_0_/G_1_ phase or G_2_/M phase, respectively. Moreover, JIB extract in combination with cisplatin diminished expression of these cell cycle-related proteins, especially Rb, p-Rb, cdk2 and cyclin D, to a greater extent than treatment with a single drug. Consequently, JIB extract plus cisplatin had a synergistic effect and mediated cell cycle progression at G_2_/M phase in OECM-1 cells.

### 2.4. Inhibitory Effect of JIB Extract Plus Cisplatin on Akt/mTOR Pathway

The former results indicated that JIB extract or combination treatment exerts antiproliferative effects, and Akt/mTOR signaling is a general pathway in tumorigenesis that plays a vital role in the regulation of cell proliferation and survival. Therefore, we aimed to elucidate whether JIB extract synergized with cisplatin to repress Akt/mTOR signaling in OECM-1 cells. JIB extract plus cisplatin enhanced the reduction in the levels of Akt, mTOR and P70S6K and their phosphorylated protein forms by approximately 2-fold (Figure 4B). In addition, NF-κB protein expression was detected, and combination treatment reduced NF-κB protein levels, while JIB extract or cisplatin alone had no marked effect on NF-κB protein levels (Figure 4B). These results revealed that JIB extract synergizes with cisplatin to inhibit cell proliferation by obstructing Akt/mTOR signaling.

### 2.5. Induction of Apoptosis in JIB Extract and Combinational Treatment

To further clarify the induction of cell death by JIB extract or combination treatment, a TUNEL assay was performed. JIB extract induced cell death by triggering OECM-1 cell apoptosis, similar to previously reported cisplatin-mediated induction of cell apoptosis. Additionally, JIB extract in combination with cisplatin induced OECM-1 cell death by activating apoptosis and DNA fragments, anoikis and apoptotic body formation were observed (Figure 5A). Next, general caspase activity after treatment was detected. Treatment with either JIB extract or cisplatin only increased general caspase activity at all time points. Combination treatment with JIB extract and cisplatin increased the caspase activity to the greatest extent compared with treatment with each drug alone (Figure 5B). These results demonstrated that JIB extract not only induced cell apoptosis, but also synergized with cisplatin to increase caspase activation.

### 2.6. Reduction of OECM-1 Cell Regrowth in JIB Extract Plus Cisplatin

During the therapeutic period for oral cancer patients, the recurrence rate is an important factor that affects prognosis. As a result, we wondered whether treatment with JIB extract in combination with cisplatin would reduce cell regrowth or prolong the inhibition of cell regrowth in vitro. JIB extract at a concentration of 30 μg/mL had a slight inhibitory effect on cell growth in OECM-1 cells grown continuously for 16 days. At the beginning of the treatment period, cisplatin had a strong inhibitory effect on OECM-1 cell growth, with an absorbance of 0.2 measured in the cell growth assay; however, after 9 and 16 days of treatment, OECM-1 cells had regrown, and an absorbance of 0.4 was recorded. When OECM-1 cells were treated with JIB extract in combination with cisplatin, combination treatment continuously inhibited cell growth, and the absorbance of cells under combination treatment was maintained below 0.2 (Figure 6). As a result, JIB extract synergized with cisplatin to mitigate the regrowth of OECM-1 cells, revealing that it may decrease recurrence after surgery or therapeutic procedures.

## 3. Discussion

With the increasing incidence of cancer, several anticancer agents are under discovery and chemically modified to improve the therapeutic treatment efficacy. Although many chemotherapeutic drugs are continuously discovered, avoiding adverse effects from chemotherapeutic drugs during treatments is difficult [19]. For instance, cisplatin has been reported to have unpredictive nephrotoxicity and immunosuppressive effects during therapy. Nonetheless, cisplatin is the first-line chemo-drug for advanced non-small cell lung cancer cells and patients with locally advanced head and neck cancers, including oral cancer, oropharyngeal cancer, hypopharyngeal cancer and laryngeal cancer [20]. Moreover, during treatment, drug resistance may develop, necessitating higher doses to achieve former antitumor effects; however, higher doses may exaggerate severe side effects [21]. Accordingly, combination treatment with various drugs with distinct mechanisms is a therapeutic strategy that may potentiate the therapeutic efficacy of each drug [22,23,24,25]. In view of these problems, combined treatment with other anticancer agents is an effective strategy to overcome those limitations for clinical application.

As a result, herein, we utilized plant extract to achieve the goal of inhibiting tumor growth and reducing cytotoxicity to alleviate possible side effects. Various natural plant extract has been studied and have developed as anticancer agents in current standard treatment, such as vincristine and paclitaxel. *Juniperus indica Bertol* has been used in daily for prevention of disease and literature search revealed few studies on *Juniperus indica Bertol* In this study, we first found that JIB extract demonstrated anticancer activity against oral cancer cells. In addition, our results indicated that JIB extract has a better selectivity index than cisplatin. These results revealed that the application of JIB extract for clinical treatment may decrease the risk of side effects in patients. The anticancer activity of JIB extract was first found to inhibit oral cancer cell growth by blocking cell cycle progression, repressing Akt/mTOR signaling and triggering cell apoptosis. Hence, JIB extract has the potential to be used as an adjuvant in combination with chemo-drugs and to promote treatment effects. Therefore, we have provided experimental evidence that the combination of JIB extract and cisplatin may enhance therapeutic potential in oral cancer treatment. In addition, JIB extract and cisplatin are regulated through inhibition of the cell cycle process, which is mediated by Rb, cdks and cyclins. Our results revealed the novel and synergistic functions of JIB extract and cisplatin, resulting in cell cycle arrest at G_2_/M phase and further reducing cell cycle-related protein expression. The findings in the study indicate that JIB extract in combination with cisplatin enhanced the inhibition of Akt/mTOR signaling to rescue cell proliferation. Additionally, the combination of JIB extract and cisplatin was more effective in killing oral cancer cells than either single treatment. The combination of JIB extract with cisplatin prevented and prolonged cell growth in vitro, suggesting that combination treatment may reduce the risk of recurrence or drug-resistant cell growth. Taken together, our results verified that JIB extract may be a safe agent to reduce the risk of side effects, making it an attractive candidate for future clinical assessment with cisplatin.

## 4. Materials and Methods

### 4.1. Cell Culture and Reagents

The immortalized OECM-1 human gingival squamous cancer cell line was developed by C.Y. Yang and Dr. C.L. Meng [26], and SAS human tongue squamous cancer cells were provided by Dr. Cheng-Chia Yu from Chung Shan Medical University (Taichung, Taiwan). SCC-25 human tongue squamous cancer cells, SVEC mouse vascular endothelial cells and MDCK canine kidney epithelial cells were obtained from American Type Culture Collection (Manassas, VA, USA) or the Bioresource Collection and Research Center (Hsinchu, Taiwan). OECM-1, SEVC and MDCK cells were cultured in DMEM with 10% FBS, 1% sodium pyruvate, 1% HEPES and 1% penicillin/streptomycin. SAS and SSC-25 cells were cultured in DMEM/F12 medium containing 10% FBS, 1% sodium pyruvate, 1% HEPES, 1% penicillin/streptomycin and 0.1% hydrocortisone. All the cell culture reagents were purchased from Gibco (Grand Island, NY, USA). The cells were cultured in 5% CO_2_ incubators at 37 °C and passaged with 0.05% trypsin-EDTA for follow-up treatment.

### 4.2. Juniperus Indica Bertol Crude Extract (JIB extract) Preparation

Crude extract from *Juniperus indica Bertol* was obtained from PHOENIX (Red Bank, NJ, USA) and extracted by steam distillation. The crude extract of the liquid oil was collected and maintained in a brown glass bottle away from light at 4 °C. For experimental application, JIB extract was initially dissolved in DMSO and applied at μg/mL concentrations.

### 4.3. MTT Proliferation Assay

Cell proliferation was determined by MTT colorimetric assay. Briefly, OECM-1, SAS and SSC-25 (5 × 10^3^ cells/100 μL/well) or MDCK and SVEC (1 × 10^4^/100 μL/well) cells were seeded in 96 well plates and treated with JIB extract (0–200 μL/mL) or cisplatin (0–50 μL/mL) for 24, 48 and 72 h as experimental groups, and cells were left untreated as a control group. Following incubation, the cells were incubated with MTT solution (500 μL/mL) for 8 h. After discarding the MTT solution, the formazan crystals were dissolved with dimethyl sulfoxide, and the absorbance at 550 nm was detected with a SpectraMax M5 microplate reader (Molecular Devices, San Jose, CA, USA). Cell viability was expressed as the percentage of treated cells over control cells. The percentage of DMSO in the wells was less than 0.2%. All the experiments were independently performed triplicated. The selective index (SI) was defined as the ratio of the IC_50_ in normal cells to the IC_50_ in tumor cells. An SI greater than 1 indicated greater selectivity for tumor cells, and an SI less than 1 indicated worse selectivity for tumor cells.

### 4.4. Synergistic Evaluation

Cells (5 × 10^3^ cells/100 μL/well) were seeded in 96 well plates for follow-up treatment as follows: synergistic treatment I: (1) JIB extract (0, 10, 20, 40, 60 and 80 μL/mL) and 3 μL/mL cisplatin or (2) JIB extract (0, 10, 20, 40, 60 and 80 μL/mL), synergistic treatment II: (1) cisplatin (0, 0.25, 0.5, 1, 2 and 4 μL/mL) and 30 μL/mL JIB extract or (2) cisplatin (0, 0.25, 0.5, 1, 2 and 4 μL/mL). All drug treatment groups were incubated with the drugs for 24, 48 and 72 h. Then, the procedure described for the MTT proliferation assay was carried out. The experiments were independently performed triplicated. Drug combinations were evaluated by the combination index (CI), which was calculated as the following equation: CI = IC_50_ of synergistic treatment I/IC_50_ of JIB extract + IC_50_ of synergistic treatment II/IC_50_ of cisplatin. CI values significantly less than 1.0 indicated synergy.

### 4.5. Cell Cycle Analysis

Cells were cultured in 10 cm dishes at a density of 2 × 10^6^ overnight and treated with the indicated drugs for 0, 6, 12, 24 and 48 h as follows: JIB extract only: 30 μL/mL JIB extract, cisplatin only: 3 μL/mL cisplatin, and synergistic treatment: 30 μL/mL JIB extract and 3 μL/mL cisplatin. PI dye was utilized to analyze the DNA content during the cell cycle. After the cells were harvested, they were incubated with PI (40 μL/mL) and RNase (100 μL/mL) at 4 °C overnight. The cell cycle distribution was analyzed with a FACSCalibur (BD, Franklin Lakes, NJ, USA), and data were analyzed by CellQuest software (version 1.2, BD, Franklin Lakes, NJ, USA). All the experiments were independently carried out triplicated.

### 4.6. Western Blot Analysis

Cells were grown in 10 cm dishes overnight and administered the following treatments for 48 h: JIB extract (30 μL/mL), cisplatin (3 μL/mL) and cisplatin plus JIB extract. Proteins were harvested and quantified by BCA assay (Thermo Fisher Scientific, Waltham, MA USA). After that, total proteins were resolved by 8–12% SDS-PAGE and transferred to PVDF membranes by electroblotting. The PVDF membranes were then blocked with skim milk and incubated with primary antibodies overnight at 4 °C. After washing, appropriate secondary antibodies were added for reaction, and horseradish peroxidase was then added for incubation at room temperature for 2 h. To visualize the protein bands, T-Pro LumiFast Plus chemiluminescence detection reagent (T-Pro Biotechnology, New Taipei County, Taiwan) was utilized with exposure to chemiluminescence light. Bands were quantified using ImageJ software (NIH, Bethesda, MD, USA). All the antibodies were purchased from Santa Cruz Biotechnology (Dallas, TX, USA). These experiments were independently achieved duplicated.

### 4.7. General Caspase Activity Assay

Caspase activity was determined using a Cell Meter Fluorometric caspase activity assay kit (AAT Bioquest, Sunnyvale, CA, USA). After cells were harvested by 0.05% trypsin, they were incubated with 500 μL of caspase activity solution for 4 h. Then, the cells were centrifuged, the supernatant was discarded, and 500 μL of assay buffer was added for flow cytometry. The data were analyzed by CellQuest (Becton Dickinson, Franklin Lakes, NJ, USA). The experiments were independently accomplished triplicated.

### 4.8. TUNEL Assay

An in situ cell death detection kit (Roche, Basel, Switzerland) was used following the manufacturer’s instructions. In brief, the cells were incubated in 10% neutral buffer formalin for 10 min and then fixed with methanol in 3% H_2_O_2_ for 10 min on culture slides. After exposure to 0.1% Triton X-100 for 2 min, 50 μL of TUNEL reaction solution was added and incubated for 2 h at 37 °C. Nuclear staining was performed using PI as a counterstain. Photographs were taken at 400× magnification and analyzed by Adobe Photoshop CS3 Extended version10.0 (Adobe, San Jose, CA, USA). The experiments were independently performed triplicated.

### 4.9. In vitro Resistance Assay

Cells were grown in 96 well plates at a density of 800 cells/100 μL and treated with JIB extract only (15 μg/mL), cisplatin only (0.75 μg/mL) or a combination of JIB extract and cisplatin for 5, 9 and 16 days. The drugs were administered every 3 days to ensure drug activity. After treatment, cells were fixed with 10% methanol for 10 min and stained with 0.1% crystal violet for 10 min. Finally, the cells were washed with PBS buffer, the colored crystals were dissolved with 10% acetic acid, and the absorbance at 590 nm was measured. The experiments were independently performed triplicated. Photographs were analyzed by Adobe Photoshop CS3 Extended version10.0 (Adobe, San Jose, CA, USA).

### 4.10. Statistical Analysis

Data are expressed as the mean ± standard deviation (SD). Statistical significance differences were determined by Student’s two-tailed *t*-test. A *p* value < 0.05 indicates statistical significance.

## Figures and Tables

**Figure 1 molecules-25-02746-f001:**
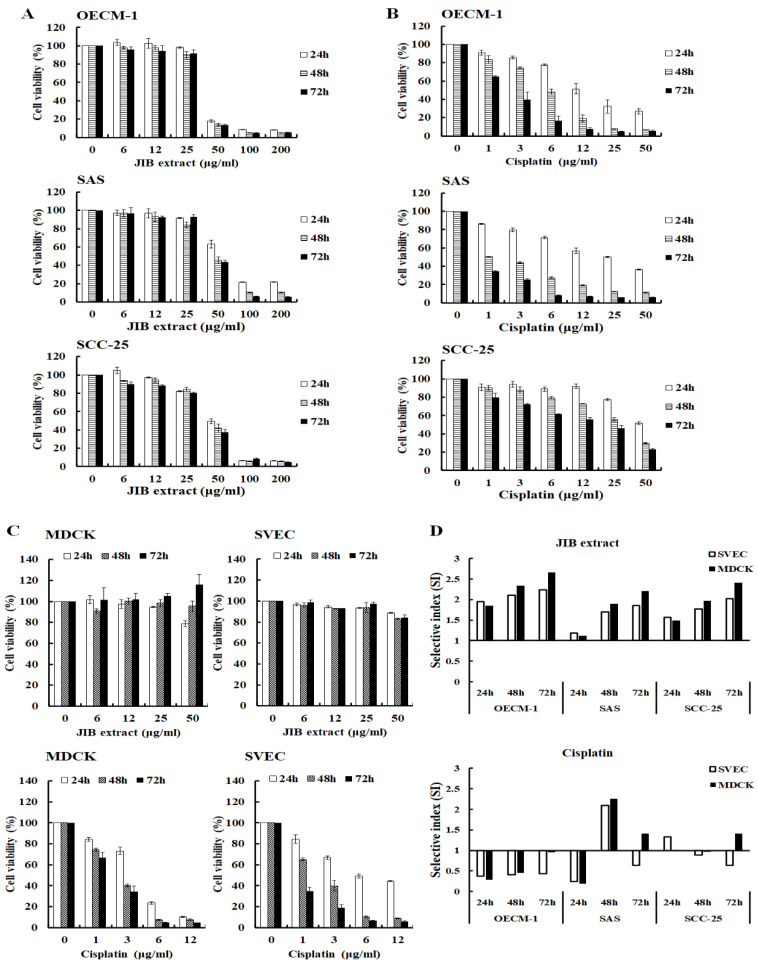
The inhibitory effect of *Juniperus indica Bertol* crude extract (JIB) on oral cancer cells. Cells were grown in 96well plates and treated with JIB extract or cisplatin at a series of concentrations for 24, 48 and 72 h. The cell viability is presented as the percentage of the mean ± SD. (**A**) Growth inhibition curves following treatment with JIB extract. (**B**) Growth inhibition curves following treatment with cisplatin. (**C**) Normal cells were treated with JIB extract and cisplatin. (**D**) Selective index (SI) of JIB extract and cisplatin in oral cancer cell lines.

**Figure 2 molecules-25-02746-f002:**
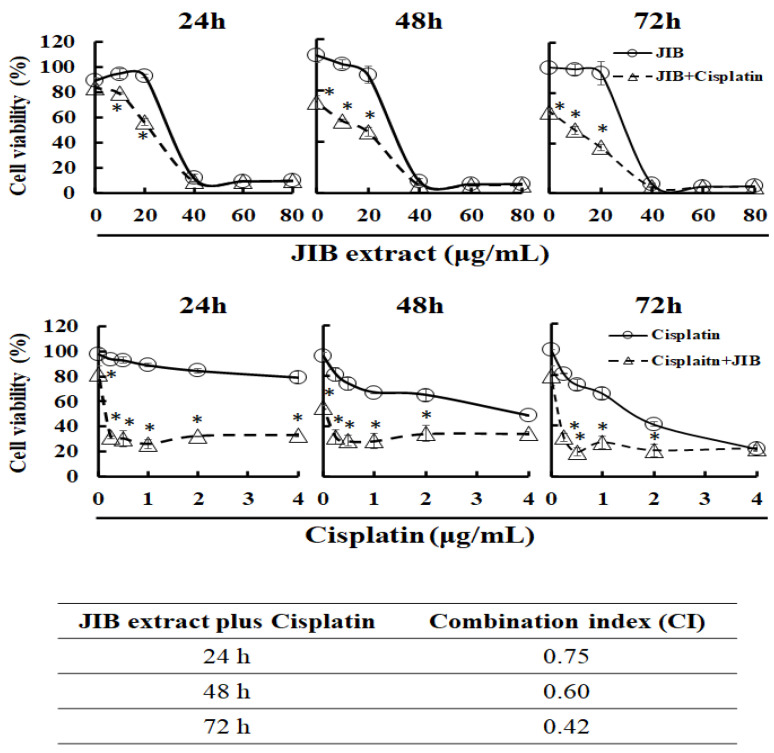
Effect of the combination of JIB extract and cisplatin on OECM-1 cells. OECM-1 cells (5 × 10^3^) were seeded in 96 well plates and simultaneously treated with JIB extract and cisplatin for 24, 48 and 72 h. Cell viability is shown as the mean ± SD, and the experiment was performed in triplicate. (*p* < 0.05).

**Figure 3 molecules-25-02746-f003:**
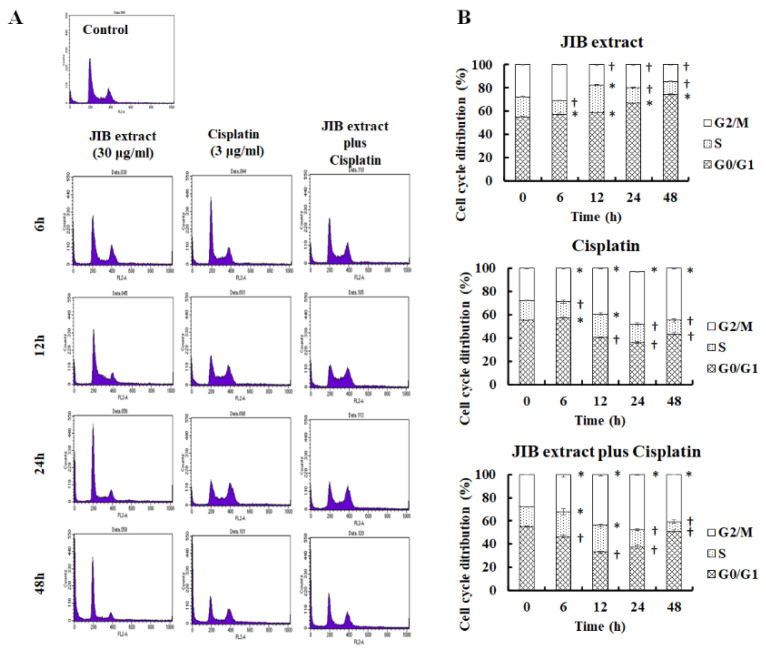
Cell cycle distribution of OECM-1 cells after treatment with JIB extract plus cisplatin. A total of 2 × 10^6^ OECM-1 were seeded in 10 cm dishes and treated with JIB extract (30 μg/mL) and cisplatin (3 μg/mL) for 6, 12, 24 and 48 h. After the cells were collected, the cell cycle was analyzed by flow cytometry, and cell cycle progression was observed. (**A**) Image showing the cell cycle distribution; (**B**) quantitative cell cycle data. *—significantly increased compared with the control; ^†^—significantly decreased compared with the control. (*p* < 0.05).

**Figure 4 molecules-25-02746-f004:**
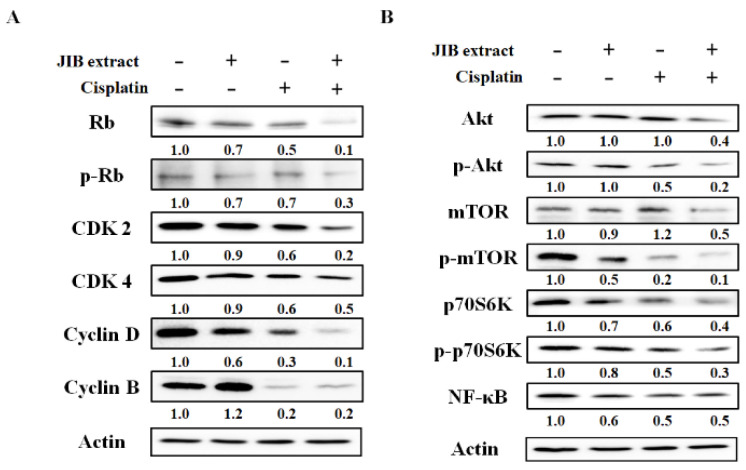
Cell proliferation-related protein expression after treatment with JIB extract plus cisplatin. After treatment, OECM-1 cells were harvested, and total protein was extracted by lysis buffer. Total proteins were quantified by BCA assay, and western blotting was performed to detect changes in protein expression due to combination treatment. (**A**) Cell cycle-related protein expression. (**B**) Akt/mTOR signaling protein expression.

**Figure 5 molecules-25-02746-f005:**
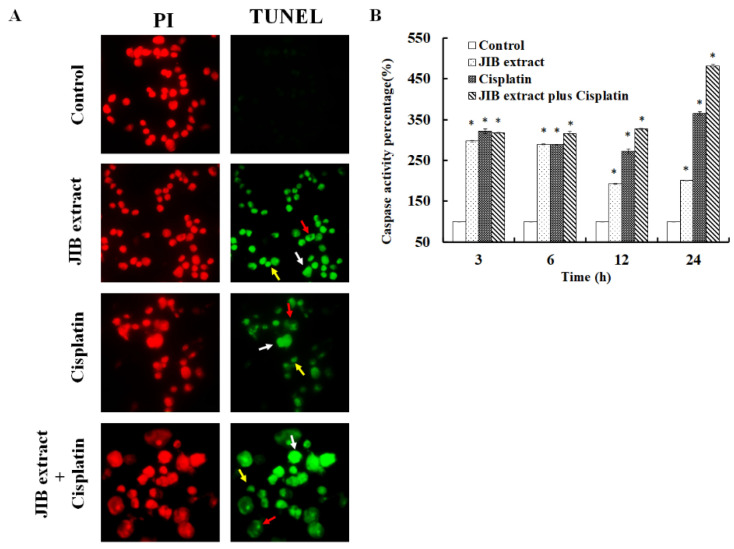
Induction of the caspase cascade after treatment with JIB extract plus cisplatin. OECM-1 cells were treated with JIB extract, cisplatin or a combination of JIB extract and cisplatin for 48 h. Then, the cells were collected, and TUNEL staining was carried out according to the manufacturer’s instructions. Red: PI; green: TUNEL-positive; white arrow: DNA fragments; red arrow: anoikis; yellow arrow: apoptotic bodies. (**A**) TUNEL staining of OECM-1 cells after drug treatment. (**B**) The caspase activity of OECM-1 cells treated after drug treatment for 3, 6, 12 and 24 h. *—significant difference between the control group and treatment groups. (*p* < 0.05).

**Figure 6 molecules-25-02746-f006:**
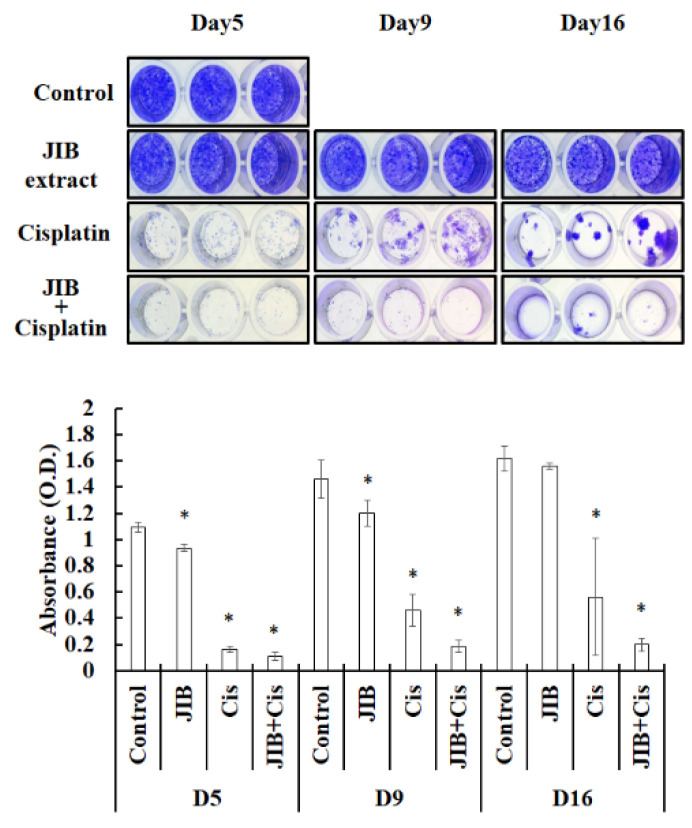
JIB extract plus cisplatin inhibited regrowth. Cells were treated with JIB extract, cisplatin or a combination of JIB extract and cisplatin for 5, 9 and 16 days, and fresh drugs were added every three days. After treatment, cells were stained with 0.1% crystal violet and photographed. The crystals were dissolved in 10% acetic acid, and the O.D. (Optical density) at 590 nm was measured. *—significant difference between the control and drug treatment groups. (*p* < 0.05).

**Table 1 molecules-25-02746-t001:** IC_50_ values of JIB extract and cisplatin in different oral cancer and normal cell lines.

Cell Line	Tumor Type	Time (h)	JIB Extract IC_50_	Cisplatin IC_50_
**Oral Cancer Cell Lines**				
OECM-1	Human oral squamous cancer cell	24 h	40 ± 0	16 ± 2
		48 h	38 ± 1	6 ± 1
		72 h	39 ± 1	3 ± 1
SAS	Human oral squamous cancer cell	24 h	66 ± 4	25 ± 2
		48 h	47 ± 2	1 ± 0
		72 h	47 ± 1	2 ± 0
SCC-25	Human tongue squamous cancer cell	24 h	50 ± 2	5 ± 0
		48 h	45 ± 2	3 ± 0
		72 h	43 ± 1	2 ± 0
**Normal Cell Lines**				
SVEC	Mouse endothelial cell	24 h	78 ± 1	6 ± 1
		48 h	80 ± 2	3 ± 0
		72 h	86 ± 4	1 ± 0
MDCK	Canine normal epithelial cell	24 h	74 ± 1	5 ± 1
		48 h	89 ± 1	3 ± 0
		72 h	103 ± 14	2 ± 0

Note: values presented as mean ± SD at different time points (μg/mL). IC_50_: 50% inhibitory concentration.

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
