# Peer review of "Extract of Juniperus indica Bertol Synergizes with Cisplatin to Inhibit Oral Cancer Cell Growth via Repression of Cell Cycle Progression and Activation of the Caspase Cascade"

_molecules, 2020, doi:10.3390/molecules25122746_

Round 1

Reviewer 1 Report

The manuscript “Extract of Juniperus indica Bertol synergizes with cisplatin to inhibit oral cancer cell growth via repression of cell cycle progression and activation of the caspase cascade” provides interesting results on the effects of a natural extract in cancer growth inhibition. In fact, the extract showed the capacity to inhibit the growth of cancer cells and, at the same time, less toxic effects on normal cells than the ones showed by cisplatin. Nevertheless, for further publication, I suggest an improvement on the language and the overall sentences construction along the manuscript.

Some examples:

The plant name should be revised along the manuscript according to Juniperus indica Bertol.

Line 26 replace “it anticancer” with “its anticancer”.

Please rewrite the last sentence of the Abstract.

Please rewrite the sentence starting at line 44, “Despite advances…”.

Please replace “In addition” in one of the sentences starting at line 52 or 53.

In line 61, change “has” to “have”.

In line 66, “another therapeutic compound” would rather be an alternative treatment/abstract, please revise.

The titles of section 2 should be written as, for instance, Inhibition of…, synergistic effect of…, etc.

Author Response

Dear Dr.

        We appreciate your expert reviewers for the constructive critiques and comments. All the suggestions have been incorporated into the manuscript and presented with high-lighted with yellow ink. Responses to the specific comments are described as follows.

  1. The plant name should be revised along the manuscript according to Juniperus indica Bertol.

Thank you for the suggestion and we have corrected into “Juniperus indica Bertol” along this manuscript.

  1. Line 26 replace “it anticancer” with “its anticancer”.

We have corrected the sentence in manuscript at line 26 with yellow ink and showed as the following: “The biofunctions of Juniperus indica Bertol, including its anticancer potential, have not been fully explored.”

  1. Please rewrite the last sentence of the Abstract.

Thank you for your suggestion and have rewritten as the following: “In this finding indicated that the JIB extract has a potential to develop anticancer agent and chemo therapeutic adjuvant for oral cancer.”

  1. Please rewrite the sentence starting at line 44, “Despite advances…”.

We have rewritten the sentence and showed as the following: “The current treatment not clearly improves survival benefits of patients due to the drug resistance, tolerance to chemo-drug toxicity, and high recurrent rate. Therefore, development of combination therapy represents a major new trend for oral cancer treatment.”.

  1. Please replace “In addition” in one of the sentences starting at line 52 or 53.

We have replaced “In addition” to “Besides” at line 53 and showed as the following: “Besides, these tumors are highly metastatic and drug resistant, resulting in noneffective treatment and poor prognosis.”

  1. In line 61, change “has” to “have”.

We have changed the “has” to have at line 61 and showed as the following: “Among these plants, the various biofunctions of Juniperus communis have been fully studied.”

  1. In line 66, “another therapeutic compound” would rather be an alternative treatment/abstract, please revise.

We replaced “another therapeutic compound” to “alternative treatment” at line 66 and showed as the following: “Hence, the purpose of the study was to investigate the anticancer potential and synergistic effect of JIB extract in combination with cisplatin to provide alternative treatment for clinicians and patients.”

  1. The titles of section 2 should be written as, for instance, Inhibition of…, synergistic effect of…, etc.

We have rewritten the titles of section 2 and have been incorporated into the manuscript.

Reviewer 2 Report

This article by Huang et al, which describes about the synergistic effect of plant extract with cis-platin in arresting cancer cell growth reads well and displays largely well structured research. Authors aim to show the benefit of combining both ingredients and succeeded to that. Elucidation of the possible mechanism of action is appreciable. However, the lack of cytotoxicity results for extract+cisplatin is noticeable.  Hence, this reviewer encourages adding cell viability data for one normal cell line and compare it with OECM cells for SI @30+3ug/ml (or any other CI).

Author Response

Dear Dr.

   We appreciate your expert reviewers for the constructive critiques and comments. The response of the question is descripted as below and we have provided the comparing Table for tumor and normal cells to help you understanding the explanation.

    We have found that the inhibition of JIB extract with highly selective index than cisplatin on OECM-1 cells, that indicated JIB extract play a major selective role in combinational experiments. Moreover, MDCK cells treating with JIB extract presented highly cell viability, ranged from 95±1% to 105±2%. As a result, we assumed that combinational treatment would inhibit normal cell growth which is less than OECM-1 cells.

Table. Comparing cell viability in tumor and normal cells.

Cell line

Treatment

24h

48h

72h

OECM-1

JIB extract (30 μg/ml)

92±10

82±1

75±3

Cisplatin (3 μg/ml)

85±2

74±1

40±8

JIB extract plus Cisplatin

33±1

27±2

21±1

MDCK

JIB extract (30 μg/ml)

95±1

98±3

105±2

Cisplatin (3 μg/ml)

73±6

40±1

34±6

Note: mean ± SD (%).

Reviewer 3 Report

Authors report their work on extract of Juniperus indica Bertol combined with cisplatin as anticancer compounds. Work is well performed regarding the evaluation of the combination as anticancer agents inc cell based assay and in determination of mechanism as anticancer agents. Weakness of the work is the fact that they use extract and there is no structural information available and thus of limited interest for further development. Also the characterisation of the extract is limited and will be difficult to reproduce the work. Due to this facts, there in a limited interest for scientific community.

There is not clear demonstration in what way biological data was obtained (only at one instance it was named that work was done in triplicate). In cell based assay it is of utmost importance to perform the same experiment as independent experiments (to ensure real reproducibility).

Instead authors report data e.g. IC50 in irrelevant accuracy, 65.9±3.93 µM should be reported as 66±4 µM.

Author Response

Dear Dr.

        We appreciate your expert reviewers for the constructive critiques and comments. All the suggestions have been incorporated into the text and presented with high-lighted with yellow ink. We have added the sentence as the following: “All the experiments were independently performed triplicated.” in the section 4: Materials and Methods to ensure real reproducibility and the IC50 values showed in Table 1 is also changed as suggestions and shows as below.

Table 1. The IC50 values of JIB extract and cisplatin in different oral cancer and normal cell lines

Cell line

Tumor type

Time(h)

JIB extract IC50

Cisplatin IC50

Oral cancer cell lines

OECM-1

Human oral squamous cancer cell

24 h

40±0

16±2

48 h

38±1

6±1

72 h

39±1

3±1

SAS

Human oral squamous cancer cell

24 h

66±4

25±2

48 h

47±2

1±0

72 h

47±1

2±0

SCC-25

Human tongue squamous cancer cell

24 h

50±2

5±0

48 h

45±2

3±0

72 h

43±1

2±0

Normal cell lines

SVEC

Mouse endothelial cell

24 h

78±1

6±1

48 h

80±2

3±0

72 h

86±4

1±0

MDCK

Canine normal epithelial cell

24 h

74±1

5±1

48 h

89±1

3±0

72 h

103±14

2±0

Note: Values are presented as the mean ± SD at different time points (μg/ml).

Round 2

Reviewer 3 Report

Authors improved manuscript according to the suggestions of the reviewers.